# Fibroblast Growth Factor 21 (FGF21) Administration Sex-Specifically Affects Blood Insulin Levels and Liver Steatosis in Obese *A^y^* Mice

**DOI:** 10.3390/cells10123440

**Published:** 2021-12-07

**Authors:** Elena Makarova, Antonina Kazantseva, Anastasia Dubinina, Elena Denisova, Tatiana Jakovleva, Natalia Balybina, Nataliya Bgatova, Konstantin Baranov, Nadezhda Bazhan

**Affiliations:** 1The Laboratory of Physiological Genetics, The Institute of Cytology and Genetics, 630090 Novosibirsk, Russia; antonyna-sh@yandex.ru (A.K.); dubinina_anastas@mail.ru (A.D.); elena_nsib@list.ru (E.D.); tatyanajakovleva@yandex.ru (T.J.); n.balybina@alumni.nsu.ru (N.B.); nataliya.bgatova@yandex.ru (N.B.); bazhan-nm@yandex.ru (N.B.); 2The Institute of Molecular and Cellular Biology, 630090 Novosibirsk, Russia; baranov@mcb.nsc.ru

**Keywords:** FGF21, *A^y^* mice, obesity, liver steatosis, sex differences, gene expression

## Abstract

FGF21 is a promising candidate for treating obesity, diabetes, and NAFLD; however, some of its pharmacological effects are sex-specific in mice with the *A^y^* mutation that evokes melanocortin receptor 4 blockade, obesity, and hepatosteatosis. This suggests that the ability of FGF21 to correct melanocortin obesity may depend on sex. This study compares FGF21 action on food intake, locomotor activity, gene expression, metabolic characteristics, and liver state in obese *A^y^* males and females. *A^y^* mice were administered FGF21 for seven days, and metabolic parameters and gene expression in different tissues were assessed. Placebo-treated females were more obese than males and had lower levels of blood insulin and liver triglycerides, and higher expression of genes for insulin signaling in the liver, white adipose tissue (WAT) and muscles, and pro-inflammatory cytokines in the liver. FGF21 administration did not affect body weight, and increased food intake, locomotor activity, expression of *Fgf21* and *Ucp1* in brown fat and genes related to lipolysis and insulin action in WAT regardless of sex; however, it decreased hyperinsulinemia and hepatic lipid accumulation and increased muscle expression of *Cpt1* and *Irs1* only in males. Thus, FGF21’s beneficial effects on metabolic disorders associated with melanocortin obesity are more pronounced in males.

## 1. Introduction

Fibroblast growth factor 21 (FGF21) is an endocrine messenger that regulates multiple aspects of energy homeostasis [1,2]. It is synthesized and secreted into the bloodstream mainly by the liver in response to various metabolic stresses and coordinates the response from adipose tissues, liver, and muscle to restore metabolic homeostasis [3]. Currently, FGF21 is considered an attractive therapeutic target to treat obesity and obesity-associated metabolic disorders, including non-alcoholic fatty liver disease (NAFLD) [1,4,5]. It was shown in preclinical models that administration of FGF21 or its mimetics and analogs reduces body weight, blood glucose levels, improves blood lipid profiles, increases insulin sensitivity [6,7,8], and protects from fatty liver disease [9,10] in obese animals. Human clinical trials have demonstrated the efficacy of some FGF21-based analogs in the reduction of body weight [8] and treatment of dyslipidemia, NAFLD, and type 2 diabetes mellitus [11,12].

The therapeutic effects of FGF21 were observed in mice with both diet-induced and genetic obesity caused by the absence of leptin (*ob/ob*) or its functional receptor (*db/db*) [6,7,13,14,15,16,17,18,19,20,21,22]. The ability of FGF21 to improve glucose and lipid metabolism in animal models of genetic obesity opens up prospects for its use for the correction of comorbidities associated with genetic forms of obesity in humans.

In humans, the most common forms of monogenic obesity are associated with mutations that disrupt the melanocortin-4 receptor (MC4R) signaling in the hypothalamus [23,24]. MC4R neurons regulate food intake, reward system, energy expenditure [25,26], glucose, and lipid metabolism [27,28]. Loss of MC4R function is accompanied by hyperphagia, development of obesity and insulin resistance [29,30], cardiovascular abnormalities [31], and fatty liver [32] in mice and humans [24,33]. The modern approach for treating melanocortin obesity is designing MC4R agonists [34]; however, MC4R agonists are only effective when upstream, but not downstream, MC4R signaling is impaired, and, in addition, they can cause side effects [34]. FGF21-based medications could be used as an alternative therapy or as part of a complex therapy to correct melanocortin obesity, but the pharmacological effect of FGF21 has hardly been studied in the models of melanocortin obesity.

Mice with the mutation Lethal yellow at the agouti locus (*A^y^*) represent a model of MC4R blockade [35]. The mutation results in ectopic expression of the agouti protein [36], which has a high affinity for MC4R [37]. Ectopic expression of agouti protein in the hypothalamus leads to blockade of MC4R and loss of MC4R function. Phenotypically, mice with the *A^y^* mutation are similar to MC4R knockout mice [29,30]. *A^y^* mice are hyperphagic and develop obesity, insulin resistance [38], and fatty liver [39] with age, so, *A^y^* mice may be used as a model of melanocortin obesity to investigate the pharmacological effects of FGF21. Previously, we investigated the effect of FGF21 administration on obese *A^y^* mice on body weight, food intake, blood hormone and metabolic characteristics, and expression of some genes in the different tissues and found sex-specific responses to this administration. In this experiment, FGF21-treated *A^y^* males had lower body weight (BW) and blood insulin concentrations than placebo-treated males, while FGF21-treated females had higher food intake (FI) and liver weight compared to females that received placebo [40]. These data indicate that FGF21 may differently affect the regulation of energy intake and expenditure, insulin sensitivity, and liver state in obese *A^y^* male and female mice and promote lipid accumulation in *A^y^* female livers. The aim of the present study is to compare the effects of exogenous FGF21 on energy intake, locomotor activity, metabolic characteristics, liver state, and expression of genes related to the regulation of energy homeostasis and glucose and lipid metabolism in obese *A^y^* males and females. Histological analysis of liver structure was made, and expression of marker genes for fibrosis, inflammation, and oxidative stress was measured to investigate FGF21 influence on the liver’s state.

## 2. Materials and Methods

### 2.1. Animals and Experimental Design

C57BL/6J-*A^y^* mice were bred in the SPF vivarium of the Institute of Cytology and Genetics. The mice were housed individually under a 13 h/11 h light/dark regime (light from 01:30 to 14:30) at an ambient temperature of 22 °C–24 °C. The mice were provided ad libitum access to water and standard pelleted Ssniff Rat/Mouse Maintenance diet received from ssniff Spezialdiäten GmbH Ferdinand-Gabriel-Weg D-59494 Soest, Germany. At the age of 35–40 weeks and body weight of 45–50 g, mice were randomly divided into control (administration of PBS) and experimental (administration of FGF21) groups. There were 7 males and 6 females in the control groups and 7 males and 6 females in the experimental groups. The mice were placed in PhenoMaster cages (TSE Systems GmbH, Berlin, Germany) and acclimated to the new cages for 8 days. Mouse fat and lean masses were assessed at the end of the period of acclimatization using EchoMRI-700 magnetic resonance analyzer (EchoMRI LLC, Houston, TX, USA). After acclimatization, recombinant mouse FGF21 (1 mg per 1 kg) dissolved in PBS or PBS alone was administered subcutaneously at the end of the light period (12:00–13:00) for 7 days. The day after the last injection, fat and lean masses were measured again, and then the animals were sacrificed by decapitation. Samples of trunk blood were collected; liver and interscapular brown adipose tissue (iBAT) were weighed. For histological analysis, the lobe of the liver was removed and fixed using a buffered 10% formalin solution (pH 7.4). The samples of the liver, visceral WAT in the paragonadal region, iBAT, and skeletal muscle were collected and snap-frozen in liquid nitrogen to evaluate gene expression. BW was measured daily, and intake of standard food and locomotor activity were monitored throughout the experimental period.

### 2.2. Histological Analysis of the Liver

Formalin-fixed and paraffin-embedded liver tissue sections were stained with hematoxylin and eosin. Light-optical studies were performed using an Axioskop 2 plus microscope (Carl Zeiss Microscopy GmbH, Jena, Germany). Digital micrographs of liver sections (*n* = 50 for each group) were analyzed using ImageJ software (National Institutes of Health, Bethesda, MD, USA). Hepatocytes were divided into 3 groups according to the degree of lipid accumulation: (1) Hepatocytes with large lipid inclusions in which the cytoplasm was filled with a large fat droplet that displaced the nucleus to the periphery of the cell; (2) hepatocytes with tightly adjacent small lipid inclusions, which were visualized as separate droplets, without displacement of the nucleus [41]; (3) hepatocytes with small-sized lipid droplets, which were located at a distance from each other. The volume densities of hepatocytes with small, medium, and large lipid inclusions were determined at ×100 magnification. The volume densities of macrophages and apoptotic bodies in the liver parenchyma were counted at ×400 magnification. A closed test system consisting of 1276 points was used for morphometry.

### 2.3. Plasma Assays and Triglyceride and Glycogen Measurements

Concentrations of insulin, leptin, and adiponectin were measured using Rat/Mouse Insulin ELISA Kit, Mouse Leptin ELISA Kit (EMD Millipore, St. Charles, MO, USA), and Mouse Adiponectin ELISA Kit (EMD Millipore, Billerica, MA, USA), respectively. Concentrations of glucose, triglycerides, and cholesterol were measured colorimetrically using Fluitest GLU, Fluitest TG, and Fluitest CHOL (Analyticon^®^ Biotechnologies AG Am Mühlenberg 10, 35,104 Lichtenfels, Germany), respectively. Concentrations of free fatty acids were measured using NEFA FS DiaSys kits (DiaSys Diagnostic Systems GmbH, Holzheim, Germany).

Liver triglycerides were measured as described by Zhao et al. [42]; liver glycogen was measured by the method of Roehrig and Allred [43].

### 2.4. Expression and Purification of Mouse FGF21

Expression and purification of mouse recombinant FGF21 were performed as previously described [22].

### 2.5. Relative Quantitation Real-Time PCR

Total RNA was isolated from tissue samples using ExtractRNA kit (Evrogen, Moscow, Russia) according to the manufacturer’s instructions. First-strand cDNA was synthesized using Moloney murine leukemia virus (MMLV) reverse transcriptase (Evrogen, Moscow, Russia) and oligo(dT) as a primer. TaqMan gene expression assays (Thermo Fisher Scientific, Waltham, MA USA), listed in Table 1, were used for relative quantitative real-time PCR with β-actin (*Actb)* as an endogenous control for the samples of muscles, liver, and hypothalamus, and *B2m* as an endogenous control for the samples of WAT and BAT. Sequence amplification and fluorescence detection were performed on an Applied Biosystems ViiA 7 Real-Time PCR System. Relative quantification was performed by the comparative threshold cycle (CT) method. 

### 2.6. Statistical Analysis

Each result is presented as an arithmetic mean ± SE for the sample size (i.e., number of mice) indicated. Repeated measures ANOVA with factors “sex” (male, female), “experiment” (PBS, FGF21 administration), and “days of experiment” (7) were used to analyze FGF21 effects on food intake (FI) and body weight (BW) loss. Three-way ANOVA with factors “sex”, “experiment”, and “day of experiment” (1–7) was used to analyze FGF21 effects on locomotor activity. Two-way ANOVA with factors “sex” and “experiment” was used to analyze FGF21 effects on weights of body, liver, fat, and iBAT, blood parameters, and gene expression with multiple comparisons using the post hoc Newman–Keuls test. The comparisons between single parameters were performed with a two-tailed Student’s *t*-test and nonparametric Mann–Whitney U test. Results were considered significant at *p* < 0.05. The STATISTICA 6 software package (StatSoft, TIBCO Software Inc., Palo Alto, CA, USA) was used for analysis.

## 3. Results

### 3.1. Effect of Exogenous FGF21 on Body Weight, Fat Mass, Food Intake, Locomotor Activity, and Hormonal and Metabolic Parameters in Obese A^y^ Male and Female Mice

Administration of PBS or FGF21 resulted in a slight decrease in BW in both males and females (Repeated measures ANOVA, *p* < 0.001, days of the experiment, Figure 1). In males, FGF21 had no effect on weight loss. In females, weight loss was lower in FGF21-treated mice compared to PBS-treated mice on days 4 and 5 of administration (Figure 1). However, these differences disappeared on day 6 and day 7 of administration.

In PBS-treated mice, FI initially decreased, possibly due to reaction to stress and then restored to normal by the end of the experiment. FGF21 administration abolished this stress anorexia from the first day of administration in females and from the third day of administration in males (Figure 1). FGF21-treated mice consumed more food than PBS-treated ones (Repeated measures ANOVA, *p* < 0.01, factor “experimental treatment”). There were no sex differences in FI in PBS-treated mice, and FGF21-treated females tended to consume more chow than FGF21-treated males (*p* < 0.078, Repeated measures ANOVA).

Male and female mice differed in locomotor activity. Males ran a greater distance than females during both the whole day and the dark phase of the day (*p* < 0.001, 3-way ANOVA, Figure 2). FGF21 administration increased locomotor activity in both males and females (*p* < 0.05, 3-way ANOVA, Figure 2). Therefore, FGF21 administration did not affect BW and increased locomotor activity and FI in obese *A^y^* mice regardless of sex.

Although males and females did not differ in BW, females had significantly less lean mass and significantly more fat (approximately 25–30%) in the body than males (Table 2). There were no sex differences in the weight of iBAT. FGF21 administration did not affect the amount of WAT and iBAT in either males or females.

There were no sex differences in the plasma concentrations of glucose, triglycerides, and free fatty acids (FFA) in mice that received PBS. FGF21 administration did not affect glucose and FFA levels and tended to increase triglyceride levels, the latter being more pronounced in males (Table 3). Two-way ANOVA revealed significant interactions between factors “sex” and “experimental treatment” in the influence on plasma levels of insulin and leptin and interaction at the level of tendency for cholesterol. In PBS-treated mice, the insulin level was significantly higher in males.

FGF21 administration reduced the plasma concentration of insulin only in males. In PBS-treated mice, the mean value of plasma leptin concentration was higher in males, and FGF21 administration differently affected leptin levels in males and females; it decreased in males and increased in females. An additional comparison of plasma cholesterol concentrations between PBS-treated males and females and between PBS- and FGF21-treated males using the Student’s test revealed significant differences. Cholesterol concentrations in females were lower than in males (*p* < 0.05), and administration of FGF21 lowered cholesterol concentrations in males (*p* < 0.05). FGF21 administration abolished sex differences in the blood levels of insulin and cholesterol observed in PBS-treated mice. Plasma concentrations of adiponectin were higher in PBS-treated females than in PBS-treated males, and FGF21 administration did not affect this parameter. The results indicate that exogenous FGF21 had a beneficial effect on hormonal and biochemical blood parameters only in males, while females were unresponsive to its action.

### 3.2. Effect of Exogenous FGF21 on Liver Fat Content

Histological analysis of the liver structure in mice treated with PBS revealed the signs of steatosis in both males and females. Hepatocytes with large lipid droplets that displaced the nucleus to the cell periphery (Figure 3A(a)), hepatocytes with tightly adjoining small lipid inclusions which were visualized as separate droplets without displacement of the nucleus (Figure 3A(b)), and hepatocytes with small-sized lipid droplets which were located at a distance from each other (Figure 3A(c)) were observed in the livers of both males and females. Accumulation of lipids in the nuclei of hepatocytes (Figure 3A(d)), the presence of hepatocytes with signs of apoptosis—heterochromatized nuclei, the presence of apoptotic bodies in the liver parenchyma (Figure 3A(e)), and infiltration of macrophages (Figure 3A(f)) were also observed in the liver of mice of both sexes. However, males had a higher proportion of hepatocytes with medium-sized lipid droplets and a lower proportion of hepatocytes with small-sized ones in their liver than females (Figure 3F).

In PBS-treated mice, the absolute and relative weight of the liver, as well as the content of fat in the liver, were significantly higher in males than in females (Table 1, Figure 3B(a,b),C,D), and hepatic glycogen stores were the same (data are not shown). These data, combined with the results for the hepatocyte lipid accumulation, indicate that males had more pronounced liver steatosis than females in the PBS treated groups. At the same time, females had more macrophages and apoptotic bodies (Figure 3E,G).

FGF21 administration did not affect glycogen content, reduced the absolute (by 20%) and relative weight of the liver, and also halved the fat content in the liver in males and had no effect on these values in females (Figure 3C,D). As a result, the sex differences in these parameters, observed in PBS-treated mice, disappeared after FGF21 administration. Administration of FGF21 was accompanied by sex-specific structural changes in the liver. In males, in contrast to females, FGF21 administration significantly decreased the number of hepatocytes with large lipid droplets (Figure 3B(a,c)), while in females, but not in males, this administration decreased the proportion of hepatocytes with small lipid droplets and increased the proportion of hepatocytes with medium lipid droplets (Figure 3B(b,d),F). FGF21 administration increased the number of apoptotic bodies in both males and females, but this increase was higher in males (more than 4 times) than in females (about 1.6 times) (Figure 3G). The data indicate that FGF21 administration improved liver health only in males.

### 3.3. Effect of Exogenous FGF21 on Gene Expression in Liver, Muscle, White and Brown Adipose Tissues, and Hypothalamus

In the liver, the mRNA levels of the following genes were assessed: *Fgf21, Ppara, Pgc1, Cpt1a, Pdk4* (activation of fatty acid oxidation); *Acaca, Acacb, Fasn* (lipogenesis); *Mttp, Apob* (lipid release from the liver with lipoproteins); *Gck, Pklr* (glycolysis); *G6pc, Pck1* (gluconeogenesis); *Insr, Irs1, Irs2, Slc2a2, Igf1, Stat5* (signal transduction of insulin and growth hormone, glucose transport); *Hif1a, Nfe2l2, Cd68, Tnfa, Il1b, Ccl3* (markers of oxidative stress and inflammation), and *Tgf-b1, Timp* (markers of fibrosis). There were no sex differences in the mRNA levels of genes related to lipogenesis, glycolysis, and gluconeogenesis in PBS-treated mice, and FGF21 administration did not affect the expression of these genes (Figure 4).

Regardless of FGF21 treatment, mRNA levels of genes related to fatty acid oxidation (*Fgf21, Cpt1a, Pdk4*), lipid transport (*Apob*), insulin and growth hormone signaling (*Insr, Irs1, Irs2, Igf1*), and pro-inflammatory activity of macrophages (*Cd68, Tnfa, Il1b, Ccl3*) were higher in females. FGF21 administration did not have a pronounced effect on the expression of any studied genes in males. In females, it stimulated gene expression of transforming growth factor beta-1 (*Tgf-b1)*, which activates fibrogenesis, and inhibited gene expression of antioxidant nuclear factor *Nfe2l2*. In addition, a significant interaction between factors “sex” and “experiment” was detected for mRNA level of inhibitor of metalloproteinases-1 (*Timp1*) that is upregulated during hepatic fibrogenesis.

Sex differences in gene expression were observed in adipose tissues of PBS-treated mice. In iBAT, mRNA levels of *Fgf21*, co-receptor for FGF21 receptor *Klb*, and FGF21 target genes (*Pparg* and *Cpt1*) were higher in females than in males (Figure 5).

In visceral WAT, mRNA levels of *Klb* and genes related to lipid metabolism (*Lipe, Slc2a4*) were also higher in females (Figure 6). Gene expression response to FGF21 administration was similar in adipose tissues of males and females. In iBAT, FGF21 administration increased the expression of *Ucp1* and *Fgf21* in both males and females. In visceral WAT, FGF21 administration had no effect on genes involved in fatty acid oxidation and thermogenesis in mitochondria and robustly increased the expression of genes involved in the regulation of fat metabolism (*Ppara*), genes encoding lipolysis enzymes ATGL (*Pnpla2*) and HSL (*Lipe*), and genes related to insulin actions (*Insr, Slc2a4*) in mice of both sexes. The only exception was the expression of the gene encoding KLB; exogenous FGF21 increased it in males and decreased it in females.

Sex differences in gene expression and gene expression response to FGF21 administration were observed in muscles (Figure 7). In PBS-treated mice, muscle mRNA levels of *Ucp3* and *Irs1* were higher in females. FGF21 administration increased mRNA levels of *Cpt1* and *Irs1* and at the level of tendency (*p* = 0.06, nonparametric Mann—Whitney U test) mRNA levels of *Insr* and *Slc2a4* only in males.

In the hypothalamus, sex differences were observed in the expression of the orexin gene *Hcrt,* which was higher in males regardless of FGF21 treatment. FGF21 administration had no effect on the expression of the studied genes, with the exception of the gene for leptin receptor (Figure 8). FGF21 administration increased the level of *Lepr* mRNA in males and decreased it in females.

## 4. Discussion

In this work, we evaluated the effectiveness of FGF21 for the correction of disorders caused by a decrease in MC4R activity. The relevance of this study is due to the fact that mutational disorders of MC4R signaling are the most common form of genetic obesity in humans, but the effect of FGF21 on metabolic disorders caused by a decrease in MC4R functions has been little studied. Loss of function of MC4R in the hypothalamus results in obesity, hyperphagia [29], reduction of sympathetic nerve activity (SNA) tone, and decrease in energy expenditure [44,45], hyperinsulinemia, and increased triglycerides accumulation in the liver [46]. *A^y^* mice were used as a model of a decrease in MC4R functions due to permanent blockade of receptors by the agouti protein [47].

The ability of FGF21 to reduce body weight in obese individuals [2] is one of its most important characteristics. Unlike other models of monogenic or diet-induced obesity and our previous results [40], exogenous FGF21 did not cause weight loss in *A^y^* mice of both sexes. FGF21 did not affect body weight and fat mass in *A^y^* mice, possibly due to the simultaneous increase in energy intake and expenditure. The same effect was observed in FGF21-treated New Zealand obese (NZO) mice, which were a model for polygenetic obesity and type 2 diabetes [48]. *A^y^* mice injected with FGF21 consumed more food than mice injected with PBS because PBS-treated mice reduced food intake under the influence of injections, while FGF21-treated mice did not. We have previously shown that *A^y^* mutation exaggerated stress anorexia in male mice [49], and our results suggest that FGF21 counteracts stress-induced anorexia in *A^y^* mice. It is not known whether FGF21 will increase food intake in the absence of stress; this issue requires further study. Perhaps, more prolonged and sparing exposure to FGF21 (chronic administration of FGF21 with a minipump, administration of FGF21-mimetics with prolonged action) would have reduced weight since its administration increased energy expenditure. This issue also requires further study.

The modes of FGF21 action on energy expenditure in *A^y^* mice are possibly specific for this model. FGF21 administration increased locomotor activity and significantly increased expression of the FGF21 gene in the iBAT in both males and females. FGF21 increases thermogenesis by acting on the central nervous system to activate sympathetic action [50]. In turn, adrenergic stimulation increases the expression of FGF21 in adipocytes, and FGF21 of adipocyte origin promotes activation of thermogenesis via autocrine action [51]. It is shown in the mouse model with deletion of UCP1-expressing adipocytes [52] that the mechanisms of FGF21’s influence on energy expenditure depend on the functional activity of BAT; FGF21 activates *Ucp1* expression and thermogenesis in the presence of *Ucp1* expressing adipocytes, and it increased physical activity in the absence of *Ucp1* expressing adipocytes. Inhibition of SNA activity underlies the decreased energy expenditure in mice with loss of functions of MC4R [45,53]. Possibly, the activating effect of FGF21 on locomotor activity in *A^y^* mice and significant elevation of *Fgf21* expression in BAT were associated with the inhibitory effect of *A^y^* mutation on the BAT activity.

Apparently, the administration of FGF21 increased energy expenditure in WAT in both *A^y^* males and females since it was accompanied by an increase in the expression of the genes for ATGL (*Pnpla2*), HSL (*Lipe*), and *Ppara* in mice of both sexes. The ATGL-PPARα pathway is involved in the metabolic activity of WAT, and lipolysis activation is accompanied by PPARa activation, increased mitochondrial and peroxisome activity, and an increase in energy expenditure [54].

Although the ability of FGF21 to reduce body weight was compromised in *A^y^* mice, FGF21 administration significantly decreased hyperinsulinemia and improved the liver state, but only in males. This suggests that FGF21 improves insulin sensitivity and lipid metabolism independently from weight loss in *A^y^* male mice. This result is in line with results obtained in mice with diet-induced obesity, NZO mice, and Siberian hamsters [48,55,56].

FGF21 reverses liver steatosis by acting on hepatic lipid metabolism and glucose uptake [10], inducing thermogenesis in BAT and WAT [57], improving insulin sensitivity in muscles and liver [58], and increasing glucose uptake by adipose tissues [48,56]. In *A^y^* mice, FGF21 increased iBAT expression of *Ucp1* and *Fgf21*, and expression of genes for the insulin receptor (*Insr*) and insulin-dependent glucose transporter Glut4 (*Slc2a4*) in WAT in both males and females, indicating enhancing of thermogenesis in BAT and glucose uptake in WAT. However, these changes were associated with improved steatosis only in males. FGF21 administration was ineffective to improve hyperinsulinemia and steatosis in *A^y^* females.

Obese *A^y^* male and female mice differed significantly in metabolic parameters and liver status initially. *A^y^* females were more resistant to the development of disorders caused by the inhibition of MC4R activity. Although *A^y^* females exhibited a greater degree of obesity than males in the present experiment, their metabolic disorders were less pronounced, as was also seen in MC4R knockout mice [44]. This higher resistance to obesity-associated disturbances was not associated with energy intake or physical activity, as females did not differ from males in FI and had lower locomotor activity than males, although female mice, including mice of the congenic line C57Bl/6J, usually have greater locomotor activity [59]. Sex differences in locomotor activity were possibly related to sex differences in the expression of *Hcrt* in the hypothalamus (it was higher in males), as the product of the gene orexin increases locomotor activity [60].

Initially, in females, signs of liver steatosis were less pronounced. The relative weight of the liver and the fat content in the liver in females were lower than in males, and triglycerides were deposited mainly in the form of small droplets, while in males, larger droplets were more prevalent. The lower fat content in female as compared to male livers may be due to greater triglyceride secretion from the liver (females had higher expression of the gene encoding apolipoprotein B), the higher triglyceride deposition in WAT, or more intensive oxidation in BAT (higher expression of *Cpt1* gene) and muscles (higher expression of *Ucp3* gene). Insulin resistance is a key factor in NAFLD pathogenesis [61,62], and better liver condition in females was associated with less pronounced insulin resistance. Females had lower blood insulin levels and higher mRNA levels of genes for insulin signal transduction in the liver (*Insr, Irs1, Irs2*), abdominal fat (*Slc2a4*), and muscles (*Irs1*). In obese *A^y^* females, estrogens and elevated blood levels of adiponectin may serve to maintain insulin sensitivity and lower lipid accumulation in the liver. Both estradiol and adiponectin increase insulin sensitivity and decrease lipid contents in the liver [63,64]. The effects of estradiol and adiponectin are realized directly on the periphery [65,66,67] or through central mechanisms, including the melanocortin system [68,69]. Since hypothalamic melanocortin regulation is disrupted in *A^y^* mice, sex differences in the severity of hepatic steatosis and insulin resistance may be associated mainly with the peripheral action of these hormones. It was shown in obese *A^y^* males that peripheral but not central administration of adiponectin decreased blood glucose and insulin levels and increased sympathetic nervous activity in BAT and expression of *Ucp1* in BAT and *Ucp3* in muscles [70].

FGF21 also may contribute to sex differences in initial metabolic status in *Ay* mice. FGF21 was shown to increase insulin sensitivity through direct actions on WAT and BAT [55]. *Fgf21* expression in the liver and BAT and *Klb* expression in WAT and BAT were higher in females than in males, which indicates a greater sensitivity of female adipose tissues to the action of FGF21.

Since insulin resistance and hepatic steatosis were less pronounced in females initially, it can be assumed that sex-specific factors (effects of estrogens, adiponectin, FGF21, as discussed above) masked the effect of exogenous FGF21. In females, administration of FGF21 had no effect on gene expression in muscles, while in males, it increased the expression of *Cpt1*, which indicates activation of fatty acid oxidation and insulin signaling genes, which indicates an increase in insulin sensitivity. Skeletal muscle is the major site of glucose uptake in the postprandial state, and insulin resistance in skeletal muscles is shown to be the primary defect in type 2 diabetes [71]. Therefore, in male mice, improvement in skeletal muscle insulin sensitivity may significantly improve whole-body insulin action, which can lead to decreased insulin levels in their blood. According to Camporez et al. [58,72], the mechanisms of the effect of estradiol in ovariectomized female mice and FGF21 in obese male mice on insulin sensitivity are the same. Both FGF21 and estradiol decrease hepatocellular and myocellular diacylglycerol content and reduce protein kinase Cϵ activation in the liver and protein kinase Cθ in skeletal muscle. The lack of the effect of FGF21 on muscle insulin sensitivity and liver triglyceride content in females may be associated with crosstalk between estradiol and FGF21 in their actions in muscles and liver.

FGF21 sex-specifically affected sensitivity to leptin. It lowered blood leptin and increased hypothalamic expression of the gene for leptin receptor (*Lepr)* in males and had opposite effects in females. Leptin helps to reduce the level of insulin in the blood through the central nervous system [73]. *A^y^* mice develop obesity-associated hyperleptinemia and leptin resistance [74]; partial restoration of leptin sensitivity in FGF21-treated male mice may contribute to lowering blood insulin levels in males.

Our results do not allow us to draw an unambiguous conclusion regarding the effect of exogenous FGF21 on liver health in *A^y^* females. In males, the administration of FGF21 was accompanied by the elimination of hepatocytes with large fat droplets, possibly as a result of apoptosis, because the number of apoptotic bodies increased significantly in males. In females, the number of hepatocytes with large droplets remained the same, but the number of hepatocytes with medium droplets increased, which may be a sign of further development of NAFLD. In addition, FGF21 administration sex-specifically affected the liver expression of genes related to inflammation and fibrosis. Initially, *A^y^* females differed from males by a large number of macrophages, increased expression of pro-inflammatory cytokines (*Cd68, Tnfa, Il1b, Cd3*), a large number of apoptotic bodies, and increased expression of gene for NFE2L2, a transcription factor that triggers the expression of the genes of proteins with antioxidant and cytoprotective properties [75]. Since PBS-treated females did not show signs of pronounced apoptosis, inflammation, and fibrosis, which is a mark of the progression of NAFLD [76], the attraction of macrophages to the liver and their activation in *A^y^* females can be considered an adaptive process aimed at recovering the structure and function of the liver. FGF21 administration increased expression of the transforming growth factor gene *Tgfb1*, which promotes the development of fibrosis [77], decreased in males and increased in females gene expression of inhibitor of metalloproteinases-1 (*Timp1*) that is upregulated during hepatic fibrogenesis [78], and decreased gene expression of the antioxidant nuclear factor, erythroid derived 2, like 2 (*Nfe2l2*). Further research is needed to determine if these changes in female liver are adaptive or indicative of progression to NAFLD and fibrosis.

Previously, preclinical studies of FGF21 action on liver steatosis were performed only in males. Our results clearly indicate that its action on liver health may depend on sex, and FGF21 administration outcomes may not benefit females. A mice computational model has shown that livers of males and females are metabolically distinct organs [79]. Sex differences exist in the prevalence, risk factors, fibrosis, and clinical outcomes of NAFLD [80], and our results highlight the need for preclinical testing of FGF21 actions on liver steatosis, not only in males, but also in females. It is not known if FGF21 is unable to improve liver health in only *A^y^* females or also in females with other models of steatosis. It is necessary to research the effect of FGF21 on hepatic steatosis in females with other forms of obesity. To answer the question of why FGF21 is not effective for treating steatosis in *A^y^* females, it is necessary to study the molecular mechanisms of interaction between FGF21 and sex steroids and their effect on carbohydrate and lipid metabolism in the liver.

A principal limitation of our study is that FGF21 actions may be different depending on species and cannot be directly translated to humans. In addition, we used mice with ectopic expression of agouti protein as a model of melanocortin obesity; however, this ectopic expression may result in blockade of melanocortin receptors, including MC1R and MC2R, not only in the brain, but also in the periphery, introducing some phenotypic features specific to this model.

## 5. Conclusions

FGF21 administration has male-specific therapeutic effect in obese *A^y^* mice. Although this administration does not decrease hyperphagia, it reduces hyperinsulinemia and improves liver health in *A^y^* male mice. This improvement is associated with signs of increasing of muscle insulin sensitivity and hypothalamic leptin sensitivity only in males. These data indicate that FGF21 is a potent candidate to correct metabolic abnormalities associated with melanocortin obesity in male individuals. In placebo-treated groups, *A^y^* females differ from males in less pronounced disorders in carbohydrate and lipid metabolism and accumulate less lipids in the liver. *A^y^* females are sensitive to FGF21 action, but FGF21 administration does not affect blood insulin level and does not improve liver state in them. Since preclinical studies of FGF21 actions on liver steatosis were previously conducted only in males, more studies are needed to evaluate the efficacy of FGF21 in improving glucose metabolism and NAFLD in obese females.

## Figures and Tables

**Figure 1 cells-10-03440-f001:**
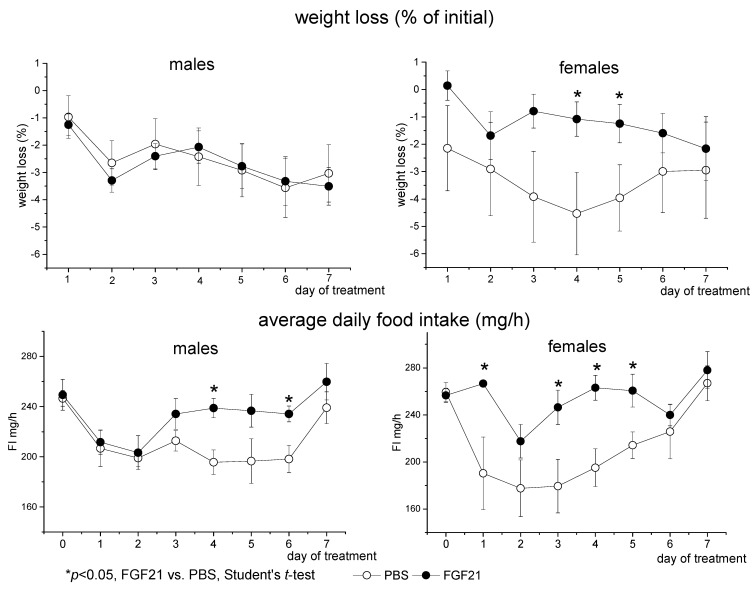
Influence of FGF21 administration on body weight and food intake in obese *A^y^* male and female mice. Data are presented as mean ± SE. Mice were administered FGF21 for 7 days from day 0.

**Figure 2 cells-10-03440-f002:**
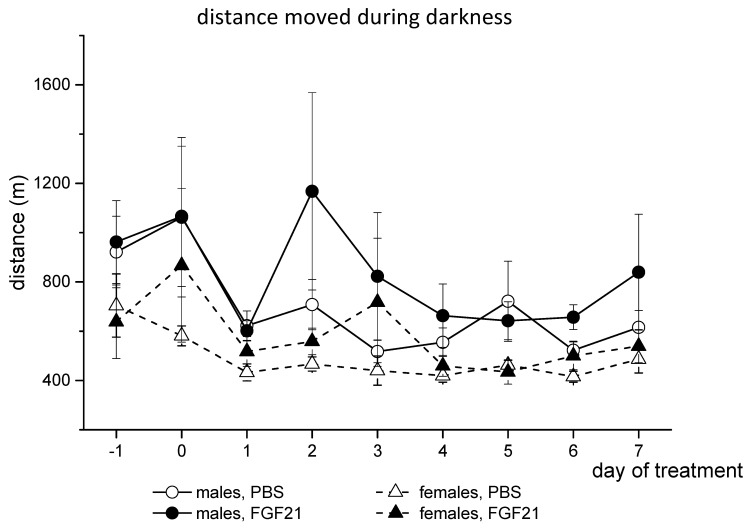
Influence of FGF21 administration on locomotor activity in obese *A^y^* male and female mice. Data are presented as mean ± SE. Mice were administered FGF21 for 7 days from day 0. Day “−1” is the day before the start of the experiment.

**Figure 3 cells-10-03440-f003:**
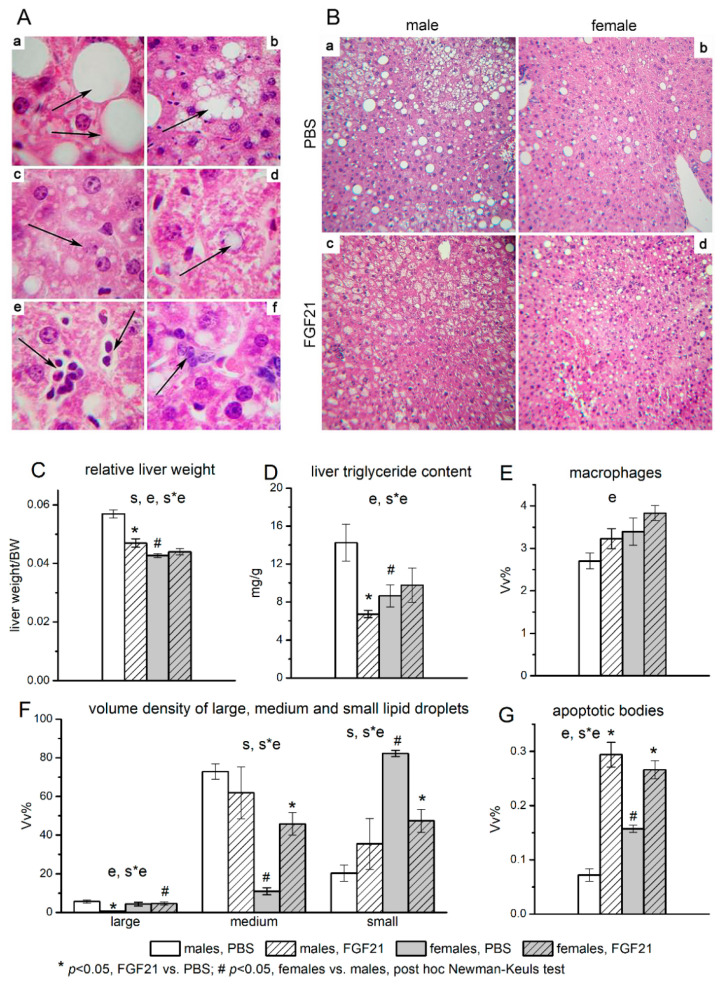
Liver structure (**A**), and influence of FGF21 administration on lipid accumulation in hepatocytes (**B**), liver index (**C**), fat content (**D**), macrophage infiltration (**E**), sizes of lipid droplets in hepatocytes (**F**), and number of apoptotic bodies (**G**) in obese *A^y^* male and female mice. A- Liver structure of C57BL/6J-*A^y^* mice. Staining with hematoxylin and eosin. Magnification ×1000. (**a**) Hepatocytes with large-size lipid inclusions occupying the entire volume of the cytoplasm (arrows). (**b**) Hepatocytes with medium-size lipid inclusions. Individual lipid droplets are visualized (arrow). (**c**) Hepatocytes with small-size lipid inclusions. Individual small lipid droplets are visible (arrow). (**d**) Lipid droplet in the hepatocyte nucleus (arrow). (**e**) Apoptotic bodies in the liver parenchyma (arrows). (**f**) Accumulation of macrophages in the liver parenchyma (arrow). B—Sex differences in lipid accumulation in the liver of C57BL/6J-*A^y^* mice. Mice were administered FGF21 for 7 days. Staining with hematoxylin and eosin. Magnification ×100. The number of hepatocytes with large and medium lipid droplets in the liver of PBS-treated males (**a**) and females (**b**) and FGF21-treated males (**c**) and females (**d**). C-G Data are presented as mean ± SE. Significant influence (*p* < 0.05) of factors “e” (administration of PBS or FGF21), “s” (sex), and “s*e”, two-way ANOVA, are indicated in the plots. The volume densities of hepatocytes with small, medium, and large lipid inclusions were determined at ×100 magnification. The volume densities of macrophages and apoptotic bodies in the liver parenchyma were counted at ×400 magnification. A closed test system consisting of 1276 points was used for morphometry.

**Figure 4 cells-10-03440-f004:**
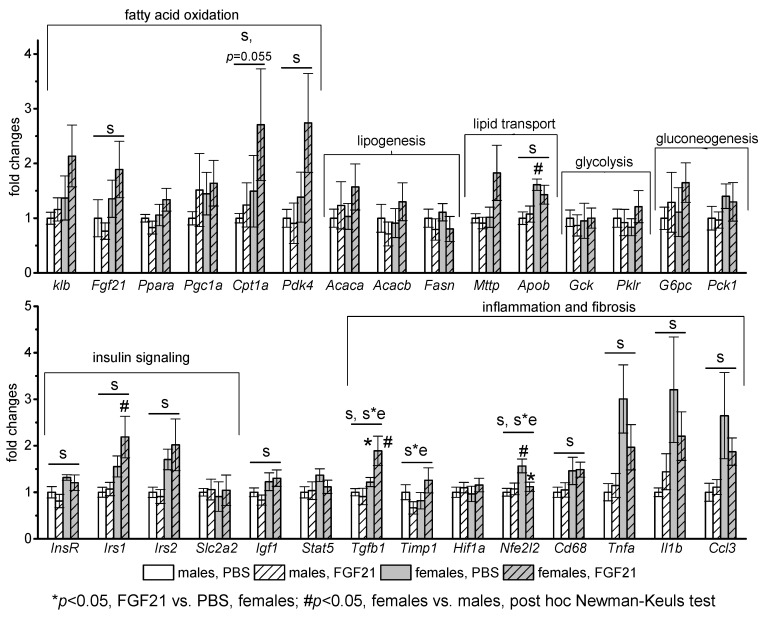
Influence of FGF21 administration on liver gene expression in obese *A^y^* male and female mice. Data are presented as mean ± SE. Mice were administered FGF21 for 7 days. Significant influence (*p* < 0.05) of factors “e” (administration of PBS or FGF21), “s” (sex), and “s*e”, two-way ANOVA, are indicated in the plot.

**Figure 5 cells-10-03440-f005:**
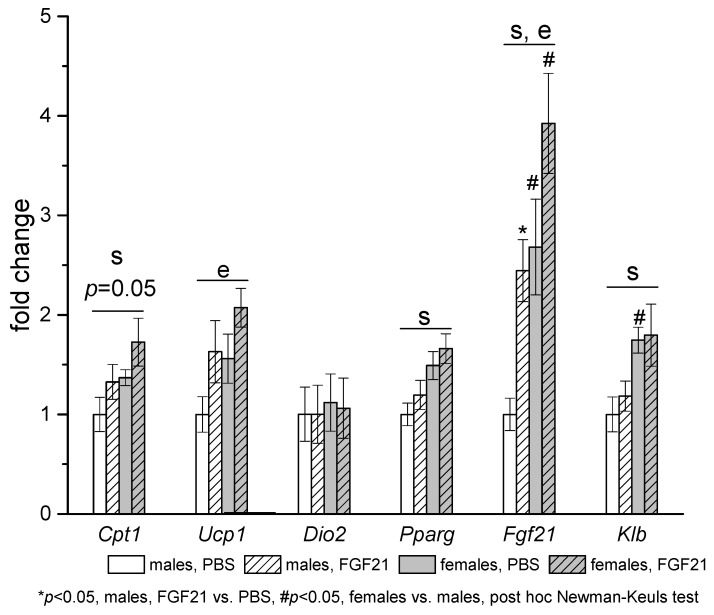
Influence of FGF21 administration on iBAT gene expression in obese *A^y^* male and female mice. Data are presented as mean ± SE. Mice were administered FGF21 for 7 days. Significant influence (*p* < 0.05) of factors “e” (administration of PBS or FGF21), “s” (sex), two-way ANOVA are indicated in the plot.

**Figure 6 cells-10-03440-f006:**
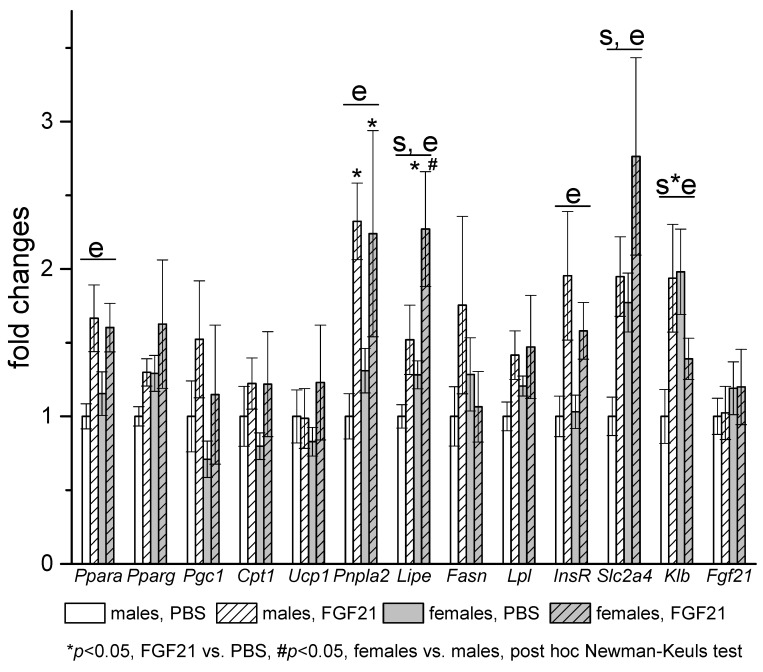
Influence of FGF21 administration on visceral WAT gene expression in obese *A^y^* male and female mice. Data are presented as mean ± SE. Mice were administered FGF21 for 7 days. Significant influence (*p* < 0.05) of factors “e” (administration of PBS or FGF21), “s” (sex), and “s*e”, two-way ANOVA are indicated in the plot.

**Figure 7 cells-10-03440-f007:**
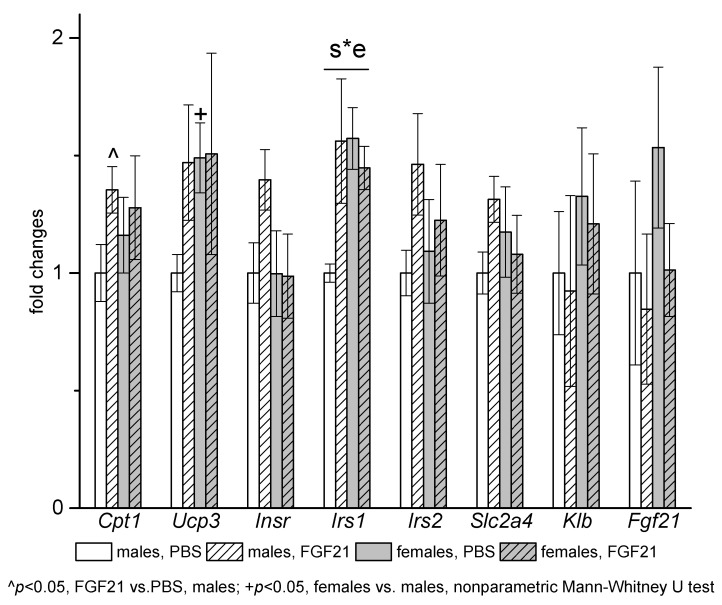
Influence of FGF21 administration on muscle gene expression in obese *A^y^* male and female mice. Data are presented as mean ± SE. Mice were administered FGF21 for 7 days. Significant influence (*p* < 0.05) of the interaction of factors “e” (administration of PBS or FGF21) and “s” (sex) “s*e” two-way ANOVA is indicated in the plot.

**Figure 8 cells-10-03440-f008:**
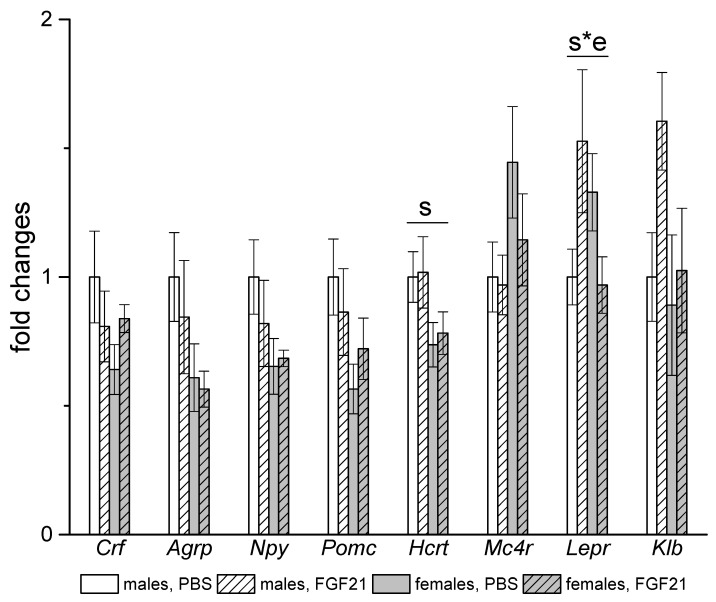
Influence of FGF21 administration on hypothalamic gene expression in obese *A^y^* male and female mice. Data are presented as mean ± SE. Mice were administered FGF21 for 7 days. Significant influence (*p* < 0.05) of factors “s” (sex) and “s*e” where “e” is the administration of PBS or FGF21, two-way ANOVA are indicated in the plot.

**Table 1 cells-10-03440-t001:** TaqMan Gene Expression Assays used for relative quantitation real-time PCR.

Protein	Gene	Gene Expression Assay ID
Acetyl-coenzyme A carboxylase alpha	*Acca*	Mm01304285_m1
Acetyl-coenzyme A carboxylase beta	*Accb*	Mm01204683_m1
Agouti related neuropeptide	*Agrp*	Mm00475829_g1
Apolipoprotein B	*Apob*	Mm01545150_m1
Beta-2-microglobulin	*B2m*	Mm00437762_m1
Beta-actin	*Actb*	Mm00607939_s1
Carnitine palmitoyltransferase 1a	*Cpt1a*	Mm01231183_m1
Carnitine palmitoyltransferase 1b	*Cpt1b*	Mm00487191_g1
CD68 antigen	*Cd68*	Mm03047343_m1
Chemokine (C-C motif) ligand 3	*Ccl3*	Mm00441259_g1
Corticotropin releasing hormone	*Crh*	Mm01293920_s1
Deiodinase, iodothyronine, type II	*Dio2*	Mm00515664_m1
Fatty acid synthase	*Fasn*	Mm00662319_m1
Fibroblast growth factor 21	*Fgf21*	Mm00840165_g1
Glucokinase	*Gck*	Mm00439129_m1
Glucose-6-phosphatase, catalytic	*G6pc*	Mm00839363_m1
Hypocretin	*Hcrt*	Mm01964030_s1
Hypoxia inducible factor 1, alpha subunit	*Hif1a*	Mm00468869_m1
Insulin receptor	*Insr*	Mm01211875_m1
Insulin receptor substrate 1	*Irs1*	Mm01278327_m1
Insulin receptor substrate 2	*Irs2*	Mm03038438_m1
Insulin-like growth factor 1	*Igf1*	Mm00439560_m1
Interleukin 1 beta	*Il1b*	Mm00434228_m1
Klotho beta	*Klb*	Mm00473122_m1
Leptin receptor	*Lepr*	Mm00440181_m1
Lipase, hormone sensitive	*Lipe*	Mm00495359_m1
Lipoprotein lipase	*Lpl*	Mm00434764_m1
Melanocortin receptor type 4	*Mc4r*	Mm00457483_s1
Microsomal triglyceride transfer protein	*Mttp*	Mm00435015_m1
Neuropeptide Y	*Npy*	Mm01410146_m1
Nuclear factor, erythroid derived 2, like 2	*Nfe2l2*	Mm00477784_m1
Patatin-like phospholipase domain containing 2 (adipose triglyceride lipase (ATGL))	*Pnpla2*	Mm00503040_m1
Peroxisome proliferative activated receptor, gamma, coactivator 1 alpha	*Ppargc1a* *(Pgc1)*	Mm01208835_m1
Peroxisome proliferator activated receptor alpha	*Ppara*	Mm0040939_m1
Peroxisome proliferator activated receptor gamma	*Pparg*	Mm00440940_m1
Phosphoenolpyruvate carboxykinase 1, cytosolic	*Pck1*	Mm01247058_m1
Pro-opiomelanocortin	*Pomc*	Mm00435874_m1
Pyruvate dehydrogenase kinase, isoenzyme 4	*Pdk4*	Mm01166879_m1
Pyruvate kinase liver and red blood cell	*Pklr*	Mm00443090_m1
Signal transducer and activator of transcription 5A	*Stat5*	Mm03053818_s1
Solute carrier family 2 (facilitated glucose transporter), member 2 (GLUT2)	*Slc2a2*	Mm00446229_m1
Solute carrier family 2 (facilitated glucose transporter), member 4 (GLUT4)	*Slc2a4*	Mm00436615_m1
Tissue inhibitor of metalloproteinase 1	*Timp1*	Mm01341361_m1
Transforming growth factor, beta 1	*Tgfb1*	Mm01178820_m1
Tumor necrosis factor alpha	*Tnfa*	Mm00443258_m1
Uncoupling protein 1 (mitochondrial, proton carrier)	*Ucp1*	Mm01244861_m1
Uncoupling protein 3 (mitochondrial, proton carrier)	*Ucp3*	Mm01163394_m1

**Table 2 cells-10-03440-t002:** Influence of FGF21 administration on the masses of the body, fat, and liver in obese *A^y^* male and female mice. Results are presented as mean ± SE for the indicated number of mice. Data were analyzed with two-way ANOVA.

	*A^y^* Males		*A^y^* Females		*p* ANOVA
	PBS (*n* = 7)	FGF21 (*n* = 7)	PBS (*n* = 6)	FGF21 (*n* = 6)	
BW (g)	46.9 ± 0.6	45.7 ± 1.2	46.5 ± 1.45	48.2 ± 0.83	ns
Lean mass (g)	28.1 ± 0.4	28.0 ± 0.3	23.8 ± 0.6 #	23.8 ± 0.5 #	s
Fat mass (g)	16.7 ± 0.6	16.0 ± 1.0	20.8 ± 1.0 #	22.5 ± 0.7 #	s
Liver weight (g)	2.67 ± 0.06	2.15 ± 0.09 *	1.98 ± 0.08 #	2,12 ± 0.06	s, e, s*e
iBAT weight (g)	0.185 ± 0.025	0.164 ± 0.017	0.179 ± 0.023	0.201 ± 0.026	ns

** p* < 0.05, FGF21 vs. PBS, males; # *p* < 0.05, females vs. males, post hoc Newman–Keuls test. Significant influence (*p* < 0.05) of factors “e” (administration of PBS or FGF21), “s” (sex), and “s*e”, or non-significant “ns”, two-way ANOVA, are indicated in the last column.

**Table 3 cells-10-03440-t003:** Influence of FGF21 administration on blood parameters in obese *A^y^* male and female mice. Results are presented as mean ± SE for the indicated number of mice. Data were analyzed with two-way ANOVA.

	*A^y^* Males		*A^y^* Females		*p* ANOVA
	PBS (*n* = 7)	FGF21 (*n* = 7)	PBS (*n* = 6)	FGF21 (*n* = 6)	
Glucose (mM)	12.6 ± 0.7	12.3 ± 1.3	10.9 ± 1.06	11.1 ± 1.22	ns
Triglycerides (mM)	0.64 ± 0.03	0.93 ± 0.13	0.71 ± 0.04	0.78 ± 0.12	*p* = 0.07, e
Cholesterol (mM)	4.29 ± 0.08	3.94 ± 0.11	3.85 ± 0.16	4.00 ± 0.12	*p* = 0.06, s*e
Free fatty acids (mM)	0.92 ± 0.11	0.64 ± 0.14	0.88 ± 0.11	0.88 ± 0.12	ns
Insulin (ng/mL)	24.2 ± 4.6	12.6 ± 1.3 *	11.0 ± 1.8 #	14.1 ± 3.4	s*e
Leptin (ng/mL)	55.6 ± 6.6	40.7 ± 5.4	46.3 ± 4.5	55.4 ± 4.0	s*e
Adiponectin (µg/mL)	8.2 ± 0.4	8.5 ± 0.2	13.0 ± 0.6 #	12.8 ± 0.5 #	s

** p* < 0.05, FGF21 vs. PBS, males; # *p* < 0.05, females vs. males, post hoc Newman–Keuls test. Significant influence (*p* < 0.05) of factors “e” (administration of PBS or FGF21), “s” (sex), and “s*e”, or non-significant “ns”, two-way ANOVA, are indicated in the last column.

## Data Availability

The data presented in this study are available on request from the corresponding author.

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
