# Peer review of "Fibroblast Growth Factor 21 (FGF21) Administration Sex-Specifically Affects Blood Insulin Levels and Liver Steatosis in Obese Ay Mice"

_cells, 2021, doi:10.3390/cells10123440_

Round 1
Reviewer 1 Report
Authors should be congratulated for approaching an interesting topic.
Major criticism
Authors should present their data as means plus/minus SD and not SEM, because readers are interested in knowing the dispersion of values not the precision of the mean, due to the paucity of observation for each group, i.e., 6/7.
Authors should enlarge the Discussion section referring to the main role of both sirtuin-4 and sirtuin-1 in this context of insulin sensitivity, insulin resistance, decreased energy expenditure and hepatic steatosis.
In fact, SIRT4 inhibits insulin secretion in pancreatic β cells and regulates insulin sensitivity as a deacylase in the pancreas. But, much more importantly, SIRT4 represses fatty acid oxidation in muscle and liver cells differently, while in WAT reduces fat oxidation and increases obesity...as evident in...The Roles of Mitochondrial SIRT4 in Cellular Metabolism. Front Endocrinol (Lausanne). 2019 Jan 7;9:783. doi: 10.3389/fendo.2018.00783. PMID: 30666234; PMCID: PMC6330279.
Recent data suggest that SIRT1 and SIRT4 were both involved in the development of insulin resistance and nonalcoholic fatty liver disease...as evident in.... Calorie restriction on insulin resistance and expression of SIRT1 and SIRT4 in rats. Biochem Cell Biol. 2010 Aug;88(4):715-22. doi: 10.1139/O10-010. PMID: 20651844.
A recent study shows low circulating levels of SIRT4 in obese patients with NAFLD mirroring its reduced mitochondrial expression in an attempt to increase the fat oxidative capacity and then the mitochondrial function in liver and in muscle. SIRT4 modulates the metabolism of free fatty acids reducing their high circulating levels but, unfortunately, increasing ROS production....as evident in...Circulating levels of sirtuin 4, a potential marker of oxidative metabolism, related to coronary artery disease in obese patients suffering from NAFLD, with normal or slightly increased liver enzymes. Oxid Med Cell Longev. 2014;2014:920676. doi: 10.1155/2014/920676. Epub 2014 Jun 17.
Coming back to another sirtuin, hepatic SIRT1 attenuates hepatic steatosis and controls energy balance in mice by inducing fibroblast growth factor 21. Gastroenterology. 2014 Feb;146(2):539-49.e7. doi: 10.1053/j.gastro.2013.10.059. Epub 2013 Nov 1. PMID: 24184811; PMCID: PMC4228483.
Finally, authors should clearly state that animal models do not completely mirror human diseases, mainly NAFLD and thus their rwsults should be taken with caution.
Author Response
We thank the Reviewer for the review and comments made.
Authors should present their data as means plus/minus SD and not SEM, because readers are interested in knowing the dispersion of values not the precision of the mean, due to the paucity of observation for each group, i.e., 6/7.
The presentation of results as mean and SEM is common. Since the number of animals in the groups was constant, then in all cases the standard error will be proportional to the standard deviation with a factor SD*1/√6 (≈SD*0.4) for females and SD*1/√7 (≈SD*0.378) for males and will proportionally reflect the variability in the groups. Based on these considerations, we left the figures in their original form. However, if the reviewer insists, we are ready to take note of his arguments.
Authors should enlarge the Discussion section referring to the main role of both sirtuin-4 and sirtuin-1 in this context of insulin sensitivity, insulin resistance, decreased energy expenditure and hepatic steatosis.
In fact, SIRT4 inhibits insulin secretion in pancreatic β cells and regulates insulin sensitivity as a deacylase in the pancreas. But, much more importantly, SIRT4 represses fatty acid oxidation in muscle and liver cells differently, while in WAT reduces fat oxidation and increases obesity...as evident in...The Roles of Mitochondrial SIRT4 in Cellular Metabolism. Front Endocrinol (Lausanne). 2019 Jan 7;9:783. doi: 10.3389/fendo.2018.00783. PMID: 30666234; PMCID: PMC6330279.
Recent data suggest that SIRT1 and SIRT4 were both involved in the development of insulin resistance and nonalcoholic fatty liver disease...as evident in.... Calorie restriction on insulin resistance and expression of SIRT1 and SIRT4 in rats. Biochem Cell Biol. 2010 Aug;88(4):715-22. doi: 10.1139/O10-010. PMID: 20651844.
A recent study shows low circulating levels of SIRT4 in obese patients with NAFLD mirroring its reduced mitochondrial expression in an attempt to increase the fat oxidative capacity and then the mitochondrial function in liver and in muscle. SIRT4 modulates the metabolism of free fatty acids reducing their high circulating levels but, unfortunately, increasing ROS production....as evident in...Circulating levels of sirtuin 4, a potential marker of oxidative metabolism, related to coronary artery disease in obese patients suffering from NAFLD, with normal or slightly increased liver enzymes. Oxid Med Cell Longev. 2014;2014:920676. doi: 10.1155/2014/920676. Epub 2014 Jun 17.
Coming back to another sirtuin, hepatic SIRT1 attenuates hepatic steatosis and controls energy balance in mice by inducing fibroblast growth factor 21. Gastroenterology. 2014 Feb;146(2):539-49.e7. doi: 10.1053/j.gastro.2013.10.059. Epub 2013 Nov 1. PMID: 24184811; PMCID: PMC4228483.
We thank the reviewer for this very useful information and will keep this in mind in our future research. We made a reference [doi: 10.1053/j.gastro.2013.10.059] to the effects of FGF21 when discussing the mechanisms of its influence on hepatic steatosis. However, we could not find any links between the action of FGF21 and SIRT 4 in the literature.
Finally, authors should clearly state that animal models do not completely mirror human diseases, mainly NAFLD and thus their rwsults should be taken with caution.
Yes, of course, we indicated this at the end of Discussion.
Reviewer 2 Report
Title: Fibroblast growth factor 21 (FGF21) administration sex-specifically affects blood insulin levels and liver steatosis in obese Ay mice
Authors: Elena Makarova, Antonina Kazantseva, Anastasia Dubinina, Elena Denisova, Tatiana Jakovleva, Natalia Balybina, Nataliya Bgatova, Konstantin Baranov and Nadezhda Bazhan
General Comment:
Despite constant progress in understanding the pathogenesis of obesity, the long-term effects of its treatment are unsatisfactory. Therefore there is a constant need for new therapeutic approaches. Data show that the fibroblast growth factor 21 (FGF21) administration can cause weight loss in genetic and diet-induced obesity. In their work, Elena Makarova et al. investigated the effects of FGF21 treatment on energy intake, locomotor activity, metabolic characteristics, liver steatosis, and expression of genes related to the regulation of energy homeostasis as well as glucose and lipid metabolism in an animal model of melanocortin obesity (Ay mice). They found that the influence of FGF21 administration on body weight, food intake, locomotor activity, and expression of some genes related to lipolysis and insulin action in brown and white adipose tissue does not depend on sex. In turn, FGF21 effects on serum insulin levels and hepatic steatosis were limited to males only. Even though obesity related to the defects of the melanocortin system is rare in humans, the study provides a plethora of descriptive data on the metabolic results of FGF21 treatment in melanocortin obesity in animals. However, some concerns regarding the manuscript's organization and structure should be addressed before it is accepted for publication.
Major revisions:
Discussion
- In the present form, the Discussion section is mainly descriptive, and apart from being quite extensive in many parts, interpretation of the observed results is limited or missing.
- Moreover, in the Discussion, the authors state, "Our results do not allow us to draw an unambiguous conclusion regarding the effect of exogenous FGF21 on liver health in Ay females". In contrast, in the Conclusions: "Female Ay mice differ from males in less pronounced disorders in carbohydrate and lipid metabolism and accumulate less lipids in the liver.” – please make these two parts of the manuscript consistent.
- Please consider and discuss possible limitations of the study and provide directions for further research.
Minor revisions:
Abstract:
- It is not clear from the abstract whether the description of Ay females refers to animals before or after FGF21 administration.
Introduction:
- “The aim of the present study was to compare the effects 77 of exogenous FGF21 on energy intake, locomotor activity, metabolic characteristics, liver state, and expression of genes related to regulation of energy homeostasis and glucose and lipid metabolism in obese Ay males and females. Influence of FGF21 administration on locomotor activity, expression of hypothalamic genes regulating energy intake and expenditure, termogenic genes in brown adipose tissue (BAT), genes related to carbohydrate and lipid metabolism in WAT, muscles and liver were assed.” – please remove the redundancy.
Material and Methods:
- Please explain why the male and female groups were not they were not of equal number?
- “Mouse fat and lean masses were assessed at the end of period of acclimatization." – please provide data on how body composition was assessed.
Whole manuscript:
- Please, reorganize reference numbers in the text according to Cells format (“… In the text, reference numbers should be placed in square brackets [ ], and placed before the punctuation; for example [1], [1–3] or [1,3].)
- Please remove double spaces within the text.
- Please explain abbreviations as only they occur in the text, e.g., "FI" & "BW."
- Table 3 and Figures 3, 4, 6, 7 & 8 – please make the table/figures content consistent with the legends and use the same abbreviations (s*e or e*s)
- The manuscript may benefit from the assistance of the native English speaker. Some of the grammar mistakes include, but are not limited to:
“It was shown in preclinical models that administration of FGF21….. and protect from fatty liver disease [9][10] in obese animals." – please consider changing to "protects."
“FGF21-based drags could serve as an alternative therapy…” – please consider changing to “drugs."
“The lock of therapeutic effect of FGF21 in female Ay mice suggests… “– please consider changing to “lack."
Author Response
We thank Reviewer for the review and remarks
Major revisions:
Discussion
- In the present form, the Discussion section is mainly descriptive, and apart from being quite extensive in many parts, interpretation of the observed results is limited or missing.
We have rewritten the Discussion to present our interpretation of the results in more detail.
- Moreover, in the Discussion, the authors state, "Our results do not allow us to draw an unambiguous conclusion regarding the effect of exogenous FGF21 on liver health in Ay females". In contrast, in the Conclusions: "Female Ay mice differ from males in less pronounced disorders in carbohydrate and lipid metabolism and accumulate less lipids in the liver.” – please make these two parts of the manuscript consistent.
It was done
- Please consider and discuss possible limitations of the study and provide directions for further research.
It was done at the end of Discussion
Minor revisions:
Abstract:
- It is not clear from the abstract whether the description of Ay females refers to animals before or after FGF21 administration.
It was improved
Introduction:
- “The aim of the present study was to compare the effects 77 of exogenous FGF21 on energy intake, locomotor activity, metabolic characteristics, liver state, and expression of genes related to regulation of energy homeostasis and glucose and lipid metabolism in obese Ay males and females. Influence of FGF21 administration on locomotor activity, expression of hypothalamic genes regulating energy intake and expenditure, termogenic genes in brown adipose tissue (BAT), genes related to carbohydrate and lipid metabolism in WAT, muscles and liver were assed.” – please remove the redundancy.
It was done
Material and Methods:
- Please explain why the male and female groups were not they were not of equal number?
We ordered the same number of males and females from the vivarium, but we lost one female when adapting to the PhenoMaster cages.
- “Mouse fat and lean masses were assessed at the end of period of acclimatization." – please provide data on how body composition was assessed.
It was done.
Whole manuscript:
- Please, reorganize reference numbers in the text according to Cells format (“… In the text, reference numbers should be placed in square brackets [ ], and placed before the punctuation; for example [1], [1–3] or [1,3].)
- Please remove double spaces within the text.
- Please explain abbreviations as only they occur in the text, e.g., "FI" & "BW."
- Table 3 and Figures 3, 4, 6, 7 & 8 – please make the table/figures content consistent with the legends and use the same abbreviations (s*e or e*s)
- The manuscript may benefit from the assistance of the native English speaker. Some of the grammar mistakes include, but are not limited to:
“It was shown in preclinical models that administration of FGF21….. and protect from fatty liver disease [9][10] in obese animals." – please consider changing to "protects."
“FGF21-based drags could serve as an alternative therapy…” – please consider changing to “drugs."
“The lock of therapeutic effect of FGF21 in female Ay mice suggests… “– please consider changing to “lack."
Manuscript was corrected by MDPI English editing service.
Round 2
Reviewer 1 Report
Manuscript improved following suggestions
Reviewer 2 Report
I want to express my gratitude for the opportunity to re-review the paper entitled: "Fibroblast growth factor 21 (FGF21) administration sex-specifically affects blood insulin levels and liver steatosis in obese Ay mice" by Elena Makarova et al. Since the authors addressed my concerns regarding the discussion construction as well as other minor revisions, I find the manuscript acceptable for publication in Cells.